# SARS-CoV-2 within-host population expansion, diversification and adaptation in zoo tigers, lions and hyenas

Laura Bashor [1], Emily N. Gallichotte[1], Michelle Galvan [1], Katelyn Erbeck[2], Lara Croft[3,4], Katelyn Stache[3], Mark D. Stenglein[1], James G. Johnson III[3], Kristy Pabilonia[2] & Sue VandeWoude [1]✉

SARS-CoV-2 rapidly adapts to new hosts following cross-species transmission; this is highly relevant as unique within-host variants have emerged following infection of susceptible wild and domestic animal species. Furthermore, SARS-CoV-2 transmission from animals (e.g., white-tailed deer, mink, domestic cats, and others) back to humans has been observed, documenting the potential of animal-derived variants to infect humans. Here, we investigate SARS-CoV-2 evolution and host-specific adaptation during an outbreak in Amur tigers (*Panthera tigris altaica*), African lions (*Panthera leo*), and spotted hyenas (*Crocuta crocuta*) at Denver Zoo in 2021. SARS-CoV-2 genomes from longitudinal samples from 16 individuals are evaluated for within-host variation and genomic signatures of selection, and we determine that the outbreak was likely initiated by a single spillover of a rare Delta sublineage. Within-host virus populations rapidly expand and diversify, and we detect signatures of purifying and positive selection, including strong positive selection in hyenas and in the nucleocapsid (*N*) gene in all animals. Four candidate species-specific adaptive mutations are identified: *N* A254V in lions and hyenas, and *ORF1a* E1724D, spike T274I, and *N* P326L in hyenas. These results reveal accelerated SARS-CoV-2 adaptation following host shifts in three non-domestic species in daily contact with humans.

The evolution of SARS-CoV-2, and the emergence of successive new variant lineages, moved to the forefront of global public health awareness during the COVID-19 pandemic. In addition to humans, the virus infects a wide range of other animal species[1–7], and it is widely accepted that SARS-CoV-2 spilled over from an animal host into human populations in late 2019[8]. Although thousands of studies have been conducted since the start of this pandemic, key questions remain unanswered—including identifying the drivers of SARS-CoV-2 evolution and transmission dynamics in non-human animals, and the origins of variant lineages infecting human and animal populations.

SARS-CoV-2 within-host evolution is shaped by both deterministic effects of selection, and stochastic effects of genetic drift. Selection acts to maximize the fitness of the population of viruses, often referred to as quasispecies, within an individual host. Limited within-host viral variation is observed in most human SARS-CoV-2 infections, with the exception of prolonged infections in immunocompromised individuals[9–12]. In these cases, a persistent infection in a single patient can result in the accumulation of a large number of selectively beneficial mutations. Similarly, virus populations can experience unique and strong selective pressures following host shifts (reviewed in ref. 13).

[1]Dept. of Microbiology, Immunology and Pathology, Colorado State University, Fort Collins, CO, USA. [2]Colorado State University Veterinary Diagnostic Laboratories, Fort Collins, CO, USA. [3]Denver Zoo Conservation Alliance, Denver, CO, USA. [4]Independent Consultant, Denver, CO, USA. ✉e-mail: sue.vandewoude@colostate.edu

**Fig. 1 | SARS-CoV-2 RNA was persistently detected in tigers, lions and hyenas by qRT-PCR followed by next-generation sequencing.** qRT-PCR was performed on nasal swabs from tigers ($N = 2$), lions ($N = 11$) and hyenas ($N = 4$) sampled repeatedly between October 2021 and January 2022. Each point represents a nasal swab sample tested for three SARS-CoV-2 gene targets, colored by the result determined by diagnostic software. All of the positive (red, $N = 68$) and inconclusive (yellow, $N = 47$) samples shown were subjected to the NGS library preparation workflow. Points with a blue diamond border indicate samples from which high-quality NGS data was successfully generated ($N = 57$ positive and $N = 6$ inconclusive samples, $N = 63$ total).

These spillover events may accelerate viral variant emergence through selection for genetic variation that improves virus fitness in a novel host environment. Capturing the initial viral dynamics following a host shift in the wild poses logistical challenges and these attributes are therefore more commonly evaluated in experimental infection systems[14,15]. In this study, we investigated SARS-CoV-2 host adaptation following a uniquely well-documented spillover event: an outbreak in large felids and hyenas living at Denver Zoo in Colorado, USA[16].

In March 2020, SARS-CoV-2 sequence analysis revealed human-to-animal transmission in five tigers and three lions at the Bronx Zoo, making non-domestic felids among the earliest documented non-human animals infected with SARS-CoV-2 through viral spillover[17,18]. Large felids have since constituted the majority of animal infections reported in zoological settings, with cases most commonly associated with respiratory signs and virus detection in oronasal samples and/or feces[16,19–32]. Furthermore, putative lion-to-human spillback has also been reported[33]. As opposed to experimental infections using model organisms, zoos offer a unique setting to understand natural infection, transmission dynamics, and viral evolution of zoonotic viruses in understudied natural hosts. Published studies most likely represent a small fraction of the true number of infections in these animals. SARS-CoV-2 infections in zoo settings offer an opportunity to evaluate risk factors and study the dynamics of variant emergence following cross-species transmission.

A detailed review of the Denver Zoo outbreak in the context of other SARS-CoV-2 outbreaks in conservatory animals has previously been reported[16]. Nasal swab samples from two Amur tigers (*Panthera tigris altaica*), eleven African lions (*Panthera leo*), and four spotted hyenas (*Crocuta crocuta*) were evaluated by SARS-CoV-2 qRT-PCR between October and December 2021, revealing variation in viral shedding among species. The findings from this comprehensive surveillance elicited further questions surrounding the exposure timeline,

transmission dynamics and potential for host adaptation. Here, we evaluate SARS-CoV-2 genomes recovered from infected animals for variant lineage, phylogenomic epidemiology, within-host variation and genomic signatures of selection. Our findings document a significant impact of host shifts on virus evolution.

## Results
### Longitudinal high-quality full genome SARS-CoV-2 sequences from zoo tigers, lions and hyenas
Nasal swab samples were collected from two tigers, eleven lions, and four hyenas over an approximately three-month period at Denver Zoo for diagnostic purposes following observation of mild respiratory signs (Fig. 1, Table S1). Detailed descriptions of all infections and complete qRT-PCR datasets have been previously described[16]. Clinical signs were observed first in tigers, followed by lions, and then hyenas. The tigers were exhibited separately ~200 m away from lions and hyenas, and supervised by different animal care specialists. Lions were housed communally in two prides and hyenas in two clans, all of which rotated through the same spaces without direct contact. Positive tests in the older hyena clan (Hyenas A and B) represent the only documented cases of SARS-CoV-2 infection in this species. Four lions had positive vRNA tests after weeks of no virus detection, including Lion B with a positive vRNA test 60 days after the first positive test in this individual.

SARS-CoV-2 vRNA was sequenced in technical duplicate to identify SARS-CoV-2 lineage, within-host variation and genomic signatures of selection. High-quality SARS-CoV-2 genome sequencing datasets were obtained from 63 biological samples from two tigers, eleven lions, and three hyenas collected at multiple timepoints between October 7th and December 13th, 2021 (Fig. 1). This corresponded to one or more timepoints for each tiger and hyena, and three to seven timepoints for each lion. Of the 68 total qRT-PCR-positive swab samples shown in Fig. 1, 57 (84%) yielded high-quality sequencing

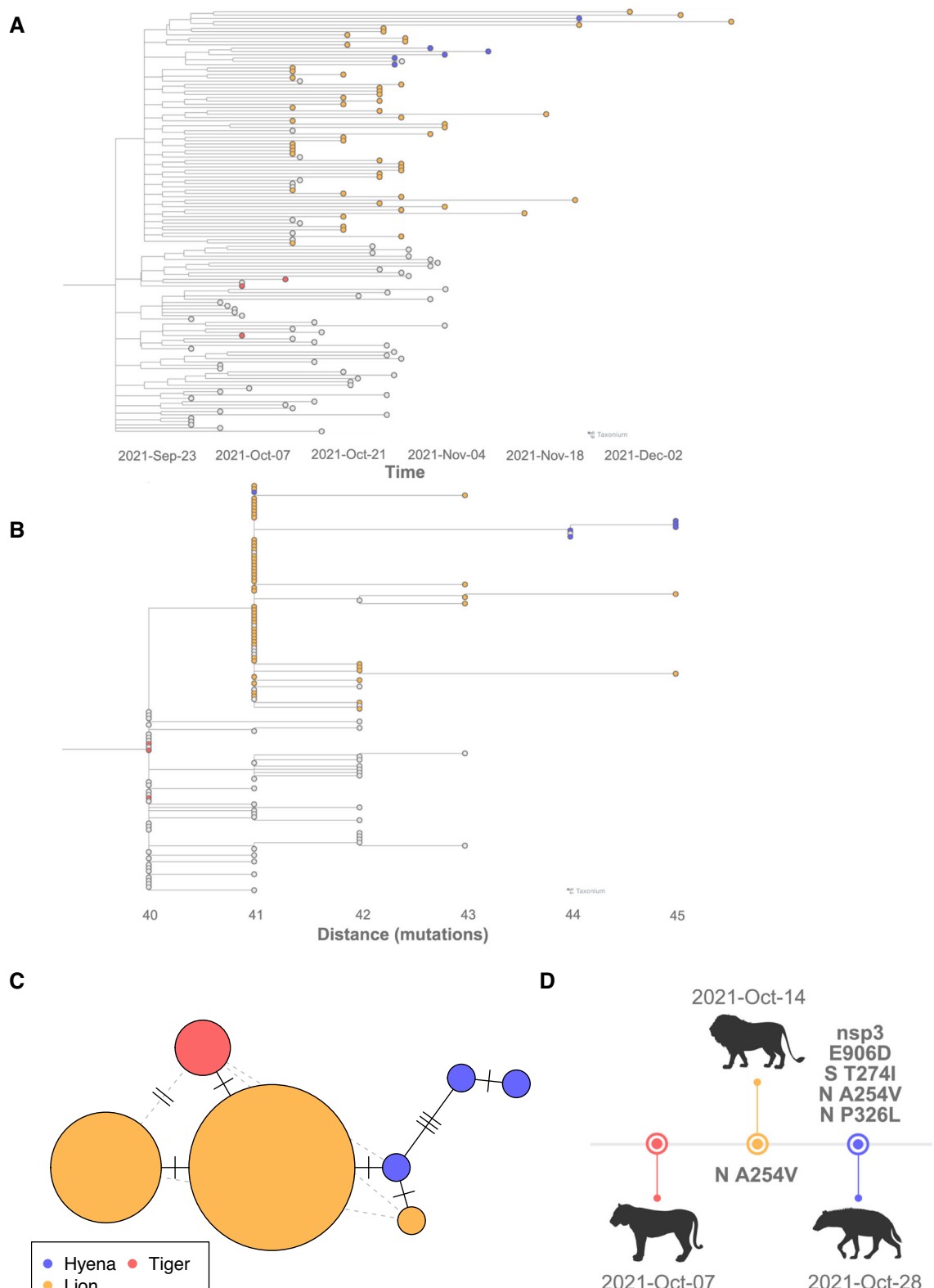

datasets. Of the 47 qRT-PCR-inconclusive samples, we acquired high-quality sequencing datasets from 6 (13%) samples.

### SARS-CoV-2 Delta sublineage AY.20 was transmitted from tigers to lions to hyenas

To obtain a focused phylogenomic epidemiology of this outbreak, a time-based tree was generated using SARS-CoV-2 sequence data from zoo animals and humans in Colorado collected during this time period (Fig. 2A, B). All sequences from animals were classified as Delta variant sublineage, Pangolin lineage AY.20, and generally clustered by species. Viruses from lions and hyenas were more closely related to each other than to those from tigers, and the two prides of lions were intermixed throughout the phylogeny. Due to limited genetic variability across all sequences, we are unable to conclusively determine if positive samples

**Fig. 2 | A single introduction of SARS-CoV-2 lineage AY.20 was transmitted from tigers to lions to hyenas.** Time-based phylogenetic tree of SARS-CoV-2 sequences from tigers, lions and hyenas, with (**A**) collection date or (**B**) divergence in number of mutations relative to the ancestral Wuhan-1 reference sequence is indicated along the x-axis. Tree was generated with Nextstrain, visualized with Taxonium, and an interactive version is available at https://nextstrain.org/community/laurabashor/DZSARS2. Data include SARS-CoV-2 consensus sequences generated from the zoo outbreak ($N = 63$), all human-derived sequences classified as AY.20 in the state of Colorado between September 23rd and November 4th,

2021 ($N = 198$), and a random subsample of human-derived sequences in Colorado from the days leading up to the zoo outbreak ($N = 1500$). Tree tips are colored by host species (tiger = red; lion = orange; hyena = blue). **C** A haplotype network of the last high-quality sequence obtained from each individual ($N = 16$; $N = 2$ tigers, $N = 11$ lions, and $N = 3$ hyenas) was generated using the 'pegas' package in R. **D** Four candidate adaptive mutations were detected in SARS-CoV-2 genomes from lions and hyenas. Timeline created in BioRender. Vandewoude, S. (2025) https://BioRender.com/boidso1.

detected in lions in November and December were recrudescent infections or reinfections; however, this homogeneity is suggestive of recrudescent infections.

The AY.20 sublineage was circulating at low rates (less than 1%) in humans globally between early 2021 and the spring of 2022, and only 0.6% of sequences in the GISAID database from Colorado between October and December 2021 were classified as AY.20[34]. Excepting twelve sequences from Denver Zoo animals generated by the United States Department of Agriculture National Veterinary Services Laboratories (USDA NVSL) following the outbreak, only three other AY.20 sequences recovered from non-human animals have been submitted to the GISAID database: two from white-tailed deer (from New York and North Carolina in November 2021), and one from a domestic cat (California in December 2021). Interestingly, these sequences group together, but were distinct from the lion, tiger and hyena genomes identified in this study (Fig. S1). This suggests that the outbreak was derived from a single introduction of AY.20 that spread among the animals at the zoo.

To further explore potential directions of transmission among animals, we generated a haplotype network (Fig. 2C). The species-specific clusters evident in this network are consistent with the observed onset of clinical signs, supporting the conclusion that SARS-CoV-2 infection began in tigers, then moved to lions, then to hyenas. Although viral genomic similarity is consistent with a single source of infection, these data do not definitively demonstrate directionality, and it is possible multiple spillover events occurred. To determine if species-specific patterns observed in this study have been replicated on a global scale, we inferred a third phylogeny from all publicly available felid- and hyena-derived SARS-CoV-2 sequences (Fig. S2). We did not observe shared mutations in this tree that would indicate felid-specific viral adaptation.

Sequences clustered due to mutations shared by more than one individual of the same species provided further evidence for the direction of transmission (Fig. 2D, Supplementary Data 1). The A254V mutation in the nucleocapsid (*N*) gene differentiated all hyena and lion sequences from three tiger-derived sequences, and was supported by robust sequencing coverage at that genomic position across all samples. SARS-CoV-2 sequences derived from Hyenas A, B, and C contained mutations P326L in N, T274I in the spike (*S*) gene, and E1724D in open reading frame (*ORF*) *1a* (nonstructural protein (*nsp*) *3* E906D). These mutations were not detected in lion or tiger samples (see methods for additional details on variant calling parameters).

### SARS-CoV-2 within-host sequence variation points to host adaptation

We identified 63 unique mutations relative to the ancestral Wuhan-1 reference sequence (Supplementary Data 1). Thirty of these were defining mutations of the Pangolin AY.20 lineage, of which the majority ($N = 27$) were nonsynonymous single nucleotide variants (SNVs) (SupS 2). The remaining 33 non-AY.20 mutations were SNVs distributed throughout the genome (11 synonymous, 12 nonsynonymous, and 10 noncoding mutations).

Based on the conclusion that the two tigers were the first to be infected, we interrogated our sequencing datasets for SARS-CoV-2 within-host variation relative to the consensus sequence from the first tiger samples (tiger reference sequence). Fourteen unique mutations remained following this approach, with limited sub-consensus

variation (Fig. 3A). The 14 unique mutations were observed 94 times in our dataset, and were almost always fixed; 14 of these observations (14.9%) were detected at < 99% allele frequency, and just five of these (5.3%) were detected at <50% allele frequency. There was more within-host variation in samples from hyenas compared to lions (Fig. 3B; lion mean = 1.33, median = 1 within-host variant per sample; hyena mean = 4.17, median = 5; Wilcoxon two-sample two-sided test, W = 273, $p = 0.000107$). The *N* A254V mutation was shared across every sample from lions and hyenas, and was the lone within-host variant detected in multiple animals (Lions B, C, D, J and Hyena D) (Fig. 3A).

The first SARS-CoV-2 sequences from Hyenas A and B (October 28th) were identical at the consensus level. All Hyena A samples share an additional mutation, *ORF1b* G2068E, which was detected sub-consensus on October 28th, and not observed at any level in the other two hyenas or any lion samples (Fig. 3A). The sequence obtained from a qRT-PCR-inconclusive sample from Hyena D was more closely related to sequences recovered from lions than to those from the older Hyenas A and B (housed separately from the younger clan) (Fig. 1).

The majority of substitutions observed in our dataset were transitions (Fig. 3C). Within-host variation was dominated by nonsynonymous SNVs, which constituted nine (64%) of the fourteen unique variants, and 87 (93%) of 94 observations of these variants (Fig. 3D). The nine nonsynonymous SNVs were located in *ORF1a* ($N = 2$ in *nsp3*, papain-like protease), *ORF1b* ($N = 2$ in *nsp15*, endoribonuclease), *S* ($N = 2$), *ORF3a* ($N = 1$) and *N* ($N = 2$). The two synonymous SNVs were both in *ORF1a* (*nsp1*, leader protein and *nsp3*), and the two noncoding variants were located at the same nucleotide position in the 5' UTR.

### Within-host virus populations were characterized by expansion, diversification, and selection

To characterize selective pressures on within-host SARS-CoV-2 populations, we calculated nucleotide diversity ($\pi$) across the virus genome at nonsynonymous and synonymous sites (Supplementary Data 5 and 6, Fig. 4A, B). In lions, which were sampled repeatedly over time, we observed a significant increase in within-host nucleotide diversity over time, accounting for repeated measures of the same individuals (Fig. 4A, Table 1). Within individual lions, synonymous nucleotide diversity also increased over time (Fig. 4C, Table 1).

Genomic signatures of positive selection ($\pi N > \pi S$) were detected in 49% (31 of 63) of virus samples, whereas 51% (32 of 63) had genomic signatures of purifying selection ($\pi S > \pi N$). Although these signatures do not conclusively determine what type of selection dominated over the course of the entire infection, they reveal the type of selection occurring within an individual at one point in time. There was no overall pattern in the detection of purifying or positive selection relative to infection timeline. Positive selection at the genome level was observed in all six samples from hyenas (Hyenas A, B and D). Differences in the strength of selection acting on SARS-CoV-2 genomes among host species was particularly evident in the final vRNA samples obtained from each individual, with positive selection in hyenas and purifying selection in tigers (Fig. 5A). Although we considered signatures of selection from all three species together to examine overall patterns during the sampling period, it is important to note that the sampling period varied among species. When nucleotide diversity was calculated separately for each SARS-CoV-2 gene, interesting patterns

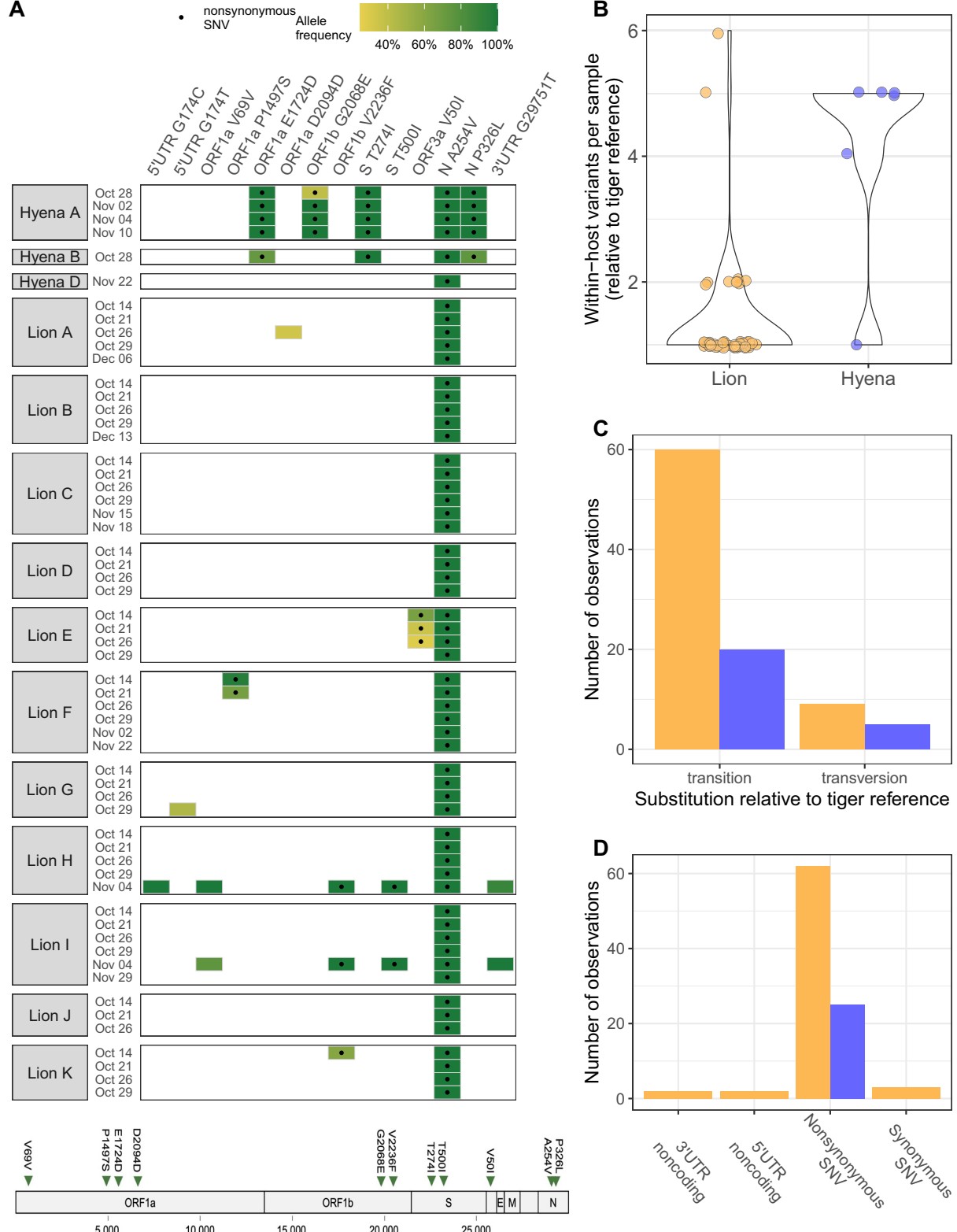

**Fig. 3 | SARS-CoV-2 within-host variation in lions and hyenas varies by species.**
**A** Fourteen within-host variants throughout the SARS-CoV-2 genome were identified in lions and hyenas (*N* = 58 samples). Each row represents one nasal swab sample, and collection date and animal identifier are indicated on the y-axis. Each tile represents a mutation detected in that sample relative to the consensus sequence of SARS-CoV-2 genomes recovered from the two infected tigers and is colored by allele frequency. Black dots indicate nonsynonymous single nucleotide variants. SARS-CoV-2 genome indicates the genomic loci of the eleven coding region mutations. Code to prepare plot was adapted from the R package 'outbreakinfo'. Number of (**B**) within-host variants (Wilcoxson two-sample two-sided test, *W* = 273, *p* = 0.000107), **C** transitions and transversions, and (**D**) predicted effects of mutation by species.

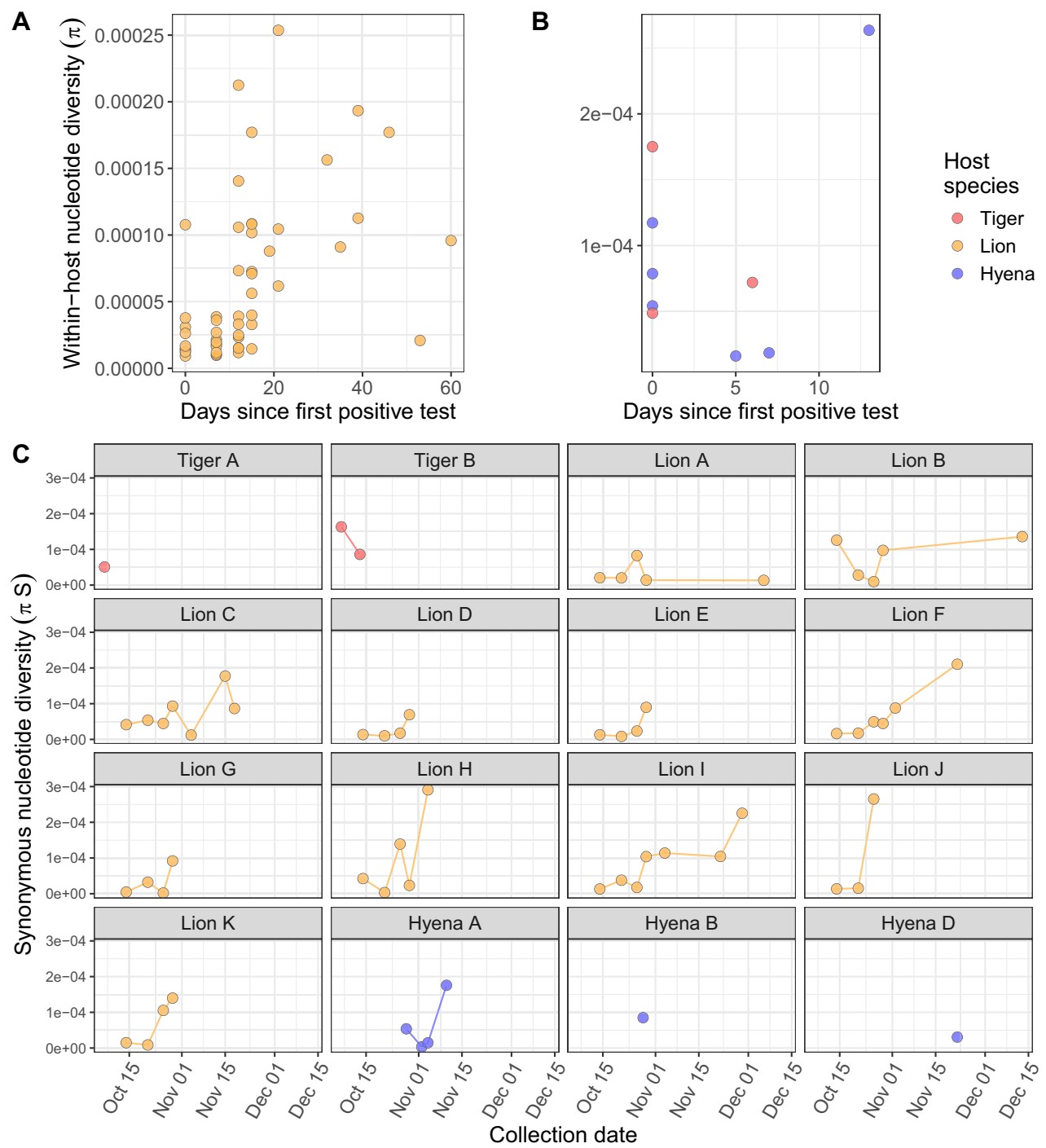

**Fig. 4 | SARS-CoV-2 within-host populations underwent expansion and diversification over time.** Each point indicates the nucleotide diversity of a SARS-CoV-2 virus population observed in one sample, calculated as the mean of two sequencing replicates. Within-host nucleotide diversity ($\pi$) by the number of days after an individual's first positive SARS-CoV-2 test, in (**A**) lions (N = 54 samples) and (**B**) tigers ($N$ = 3 samples) and hyenas ($N$ = 6 samples). **C** Within-host synonymous nucleotide diversity ($\pi S$) over time by individual animal.

of selection were observed. Strong positive selection was detected in the *N* gene from all samples across all timepoints, and most *S* genes across all species, whereas both positive and purifying selection were detected in *ORF1ab* (Fig. 5B, C). The remaining gene segments showed varied or no signatures of selection (Fig. 5C, D).

## Discussion

The challenge of capturing initial transmission dynamics and host shifts that shape pathogen emergence remains a key research question. This limitation is not exclusive to SARS-CoV-2, and remains highly relevant to other emerging infectious diseases such as highly pathogenic avian influenza, which has recently been reported in domestic livestock and cats in addition to poultry and wild birds[35]. In this study we used targeted NGS to interrogate longitudinal nasal swab samples from a SARS-CoV-2 outbreak in a zoological setting. The resulting dataset presents an exceptional opportunity for investigating virus within-host evolution across multiple species following serial cross-species transmission events. Our analysis indicates that a single

**Table 1 | Summary of linear mixed models of nucleotide diversity (π) and synonymous nucleotide diversity (πS) of SARS-CoV-2 in lions as a function of time (π model p = 0.000088; πS model p = 0.00017)**

| Predictors | Nucleotide diversity | | Synonymous nucleotide diversity | |
|---|---|---|---|---|
| | Estimates | p | Estimates | p |
| (Intercept) | 0.000031 ** | 0.006 | 0.000031 * | 0.012 |
| Time (days after first positive test) | 0.000002 *** | <0.001 | 0.000003 *** | <0.001 |
| Random Effects | | | | |
| σ² | 0.000000002600 | | 0.000000003679 | |
| τ₀₀ | 0.000000000210 animal_ID | | 0.000000000000 animal_ID | |
| ICC | 0.074647622901 | | | |
| N | 11 animal_ID | | 11 animal_ID | |
| Observations | 54 | | 54 | |
| Marginal R² / Conditional R² | 0.249 / 0.305 | | 0.236 / NA | |

*p < 0.05 **p < 0.01 ***p < 0.001

Time was measured as days since first positive qRT-PCR test. Time was a fixed effect, with animal ID as a random effect to account for repeated measures of the same individual (N = 54 samples from 11 lions) over time. The models were fit with the lmer() function in the 'lme4' package, with the 'lmerTest' package used to obtain p-values based on Satterthwaite's degrees of freedom approximation (two-sided). The table was generated with the tab_model() function in the 'sjPlot' package in R.

introduction of a rare SARS-CoV-2 Delta sublineage likely spread from humans to tigers to lions to hyenas. Repeated sampling of individual animals provided snapshots of SARS-CoV-2 within-host population expansion and diversification within hosts. Signatures of both purifying and positive selection were evident across the virus genome, along with four apparently species-specific adaptive mutations in lions and hyenas. While no known concerning SARS-CoV-2 variant lineage evolved in infected animals, uniquely strong signatures of positive selection detected in the N gene and in samples from hyenas highlight the combined impacts of mutation and selection following virus host species shifts.

The four candidate species-specific mutations observed in lions and hyenas have rarely been detected in humans and are not associated with any particular variant lineage. At the time of this writing, N A254V has been reported in just 494 (0.003%) out of the more than 15 million SARS-CoV-2 genome sequences from humans in the GISAID database[34]. The three hyena-specific mutations are similarly rare (0.01-0.3%), and have also been detected in a handful of other non-human animals including a mink, multiple white-tailed deer, a cat and an Amur tiger[34,36]. Interestingly, ORF1a E1724D and S T274I were reported in different deer sequences submitted to GISAID on the same date by the same laboratory, suggesting both mutations may have been present in the same population of deer in New York.

All virus samples collected during this outbreak were classified as the same Delta lineage AY.20. This outbreak occurred during one of the peaks of the COVID-19 pandemic, but less than 1% of human infections in Colorado at this time were the AY.20 sublineage. The rarity of the lineage among humans around this time adds further support to the hypothesis that the zoo outbreak was initiated by a single spillover event, presumably from an infected human, although it is possible that multiple independent spillovers from humans into animals occurred, or that transmission occurred among peridomestic animals at the zoo. In the GISAID database, there are 17 total SARS-CoV-2 sequences recovered from humans classified as AY.20 with the N A254V mutation (detected in our study in lions and hyenas, but not tigers)[34]. One was collected in Mexico in September 2021, and the remaining sixteen were collected between October 14th and 29th, 2021 in Colorado, nearly coincident with the diagnosis of infection in Denver Zoo lions (Fig. 6). While this putative lion-derived AY.20 lineage thus apparently retained the ability to infect humans, it also rapidly disappeared from the human population. Furthermore, the relative transmissibility of this lineage (among animals and between animals and humans)—or that of a lineage carrying the candidate hyena-adapted mutations—remains undetermined. These findings point to

the importance of employing public health strategies that are informed by One Health principles, taking into account animals, human and the environment. Surveillance of confirmed or potential wildlife hosts is important, but often limited by logistics and costs. This study demonstrates the valuable context that can be gained from zoological collections, including the targeted surveillance of vulnerable populations of animals, and analyses of within-host viral evolution[16].

When we narrowed the focus of our analysis to within-host viral variation in lions and hyenas relative to the tiger index cases, we observed the unique trajectory of virus populations within individual hosts (Fig. 3). Although the N A254V mutation was fixed in all lions and hyenas from the first samples onwards, it was not observed in any tiger samples despite technical replication and robust sequencing depth. This suggests the mutation emerged in animals during the outbreak and was maintained by positive selection. Viruses recovered from hyenas had more genetic variation than those from lions, including the three potentially adaptive mutations ORF1a E1724D, S T274I, and N P326L.

The strongest signatures of positive selection were observed in the SARS-CoV-2 N gene which encodes the RNA-binding nucleocapsid protein, and plays a crucial structural role in packaging the viral genome (Fig. 5). This is not surprising, as mutations in the N gene have been associated with the evolution of SARS-CoV-2 variants of concern in humans; most notably, R203K/G204R have been shown to increase viral replication and fitness[37,38]. Both A254V and P326L are located in the dimerization domain of N. A recent study found that introducing the P326L mutation did not destabilize the nucleocapsid dimer, and by one measurement, was more stable than the wildtype structure[39]. Increased stability of the nucleocapsid protein could improve virion survival in the host or environment, which would directly enhance SARS-CoV-2 transmissibility. In a subset of samples there were also elevated signatures of positive selection in the spike gene. The hyena-specific T274I mutation in this gene could represent adaptation to the hyena ortholog of angiotensin-converting enzyme 2 (ACE2), which mediates host cell entry. A recent review examining structural impacts of S mutations suggested that mutation at residue T274 could also contribute to CD8 + T-cell immune escape in humans, indicating some immunologic basis for selection at this site[40].

The SARS-CoV-2 variation in lions and hyenas described in this study highlights the plasticity of within-host virus populations experiencing selection and genetic drift over time. The limited sub-consensus level variation in lions is comparable to what is seen in most human infections[9], whereas hyenas bear a stronger resemblance to the increased SARS-CoV-2 within-host variation seen in chronic infections

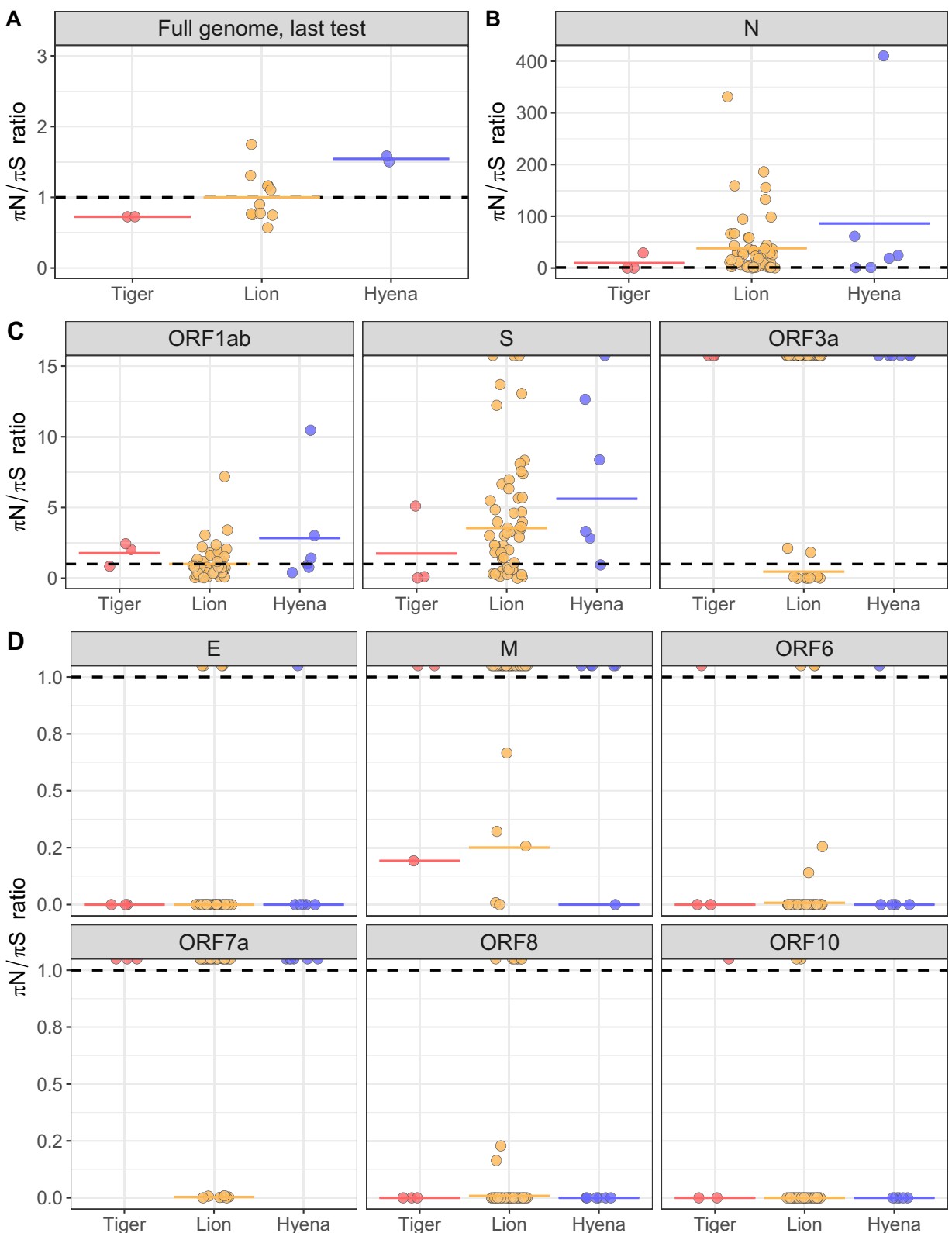

of immunocompromised humans[10–12]. The predominance of transitions over transversions in all species is also consistent with previous reports, and may be associated with apolipoprotein B mRNA-editing enzyme catalytic polypeptide (APOBEC) editing of viral genomes[41,42]. Virus populations can face substantial challenges to the maintenance of genetic variation due to transmission bottlenecks. For example, in humans SARS-CoV-2 infections are often initiated with just 1-2 virions[43], and this phenomenon likely played a role during the zoo outbreak to limit the size of the virus population transmitted from one individual to another. While the synonymous changes observed in this study do not alter viral antigen amino acid composition, such changes can have important implications for viral evolution by resulting in alternating codon usage, RNA secondary structure, innate immune evasion, and other primary and secondary effects[43–46].

**Fig. 5 | Strong signatures of positive selection were detected across species and SARS-CoV-2 genome segments.** **A** Full SARS-CoV-2 genomes from hyenas were under positive selection (πN/πS ratio >1). **B** Positive selection was observed in the nucleocapsid (N) gene across all species. **C** Mixed signatures of positive and purifying selection were detected in *ORF1ab*, *S* and *ORF3a*, and (**D**) no or exclusively purifying signatures of selection were detected in the remaining gene segments. The ratio of nonsynonymous (πN) to synonymous (πS) nucleotide diversity was calculated as a measure of the type and strength of selection acting on SARS-CoV-2 populations. Each point represents the πN/πS ratio of (**A**) a SARS-CoV-2 virus population observed in the last nasal swab sample collected for each individual

(*N* = 16 animals; *N* = 2 tigers, *N* = 11 lions, and *N* = 3 hyenas), or (**B**, **C**, **D**) a SARS-CoV-2 gene segment observed in one sample calculated as the mean of two technical sequencing replicates (*N* = 63 samples total; *N* = 54 samples from 11 lions, *N* = 3 samples from two tigers and *N* = 6 samples from three hyenas). Black dashed lines indicate πN/πS ratio of 1, and colored lines indicate the mean for each species. Points that are not fully visible represent samples for which πS = 0, which results in infinite πN/πS. These points are indicated with half points along plot borders and were not used in mean calculations. The abbreviated gene segment is indicated at the top of each plot (nucleocapsid (*N*), open reading frame (*ORF*), spike (*S*), envelope (*E*), membrane (*M*)).

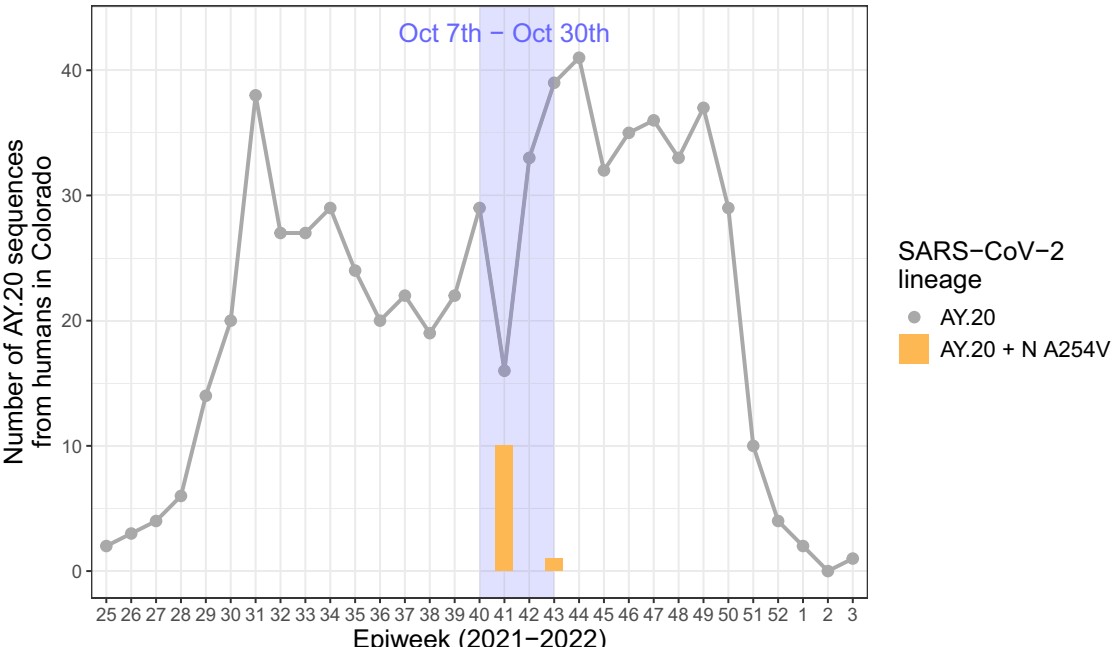

**Fig. 6 | The SARS-CoV-2 AY.20 lineage with the *N* A254V mutation (detected in lions and hyenas) was found in humans in Colorado during the early stage of the zoo outbreak.** Each point indicates the number of SARS-CoV-2 sequences in Colorado classified as AY.20 per epidemiological week (epiweek), from the first observation of AY.20 on June 23rd, 2021 (2021 epiweek 25) to the last observation on

January 16th, 2022 (2022 epiweek 3). Orange bars represent the number of SARS-CoV-2 sequences classified as AY.20 with the *N* A254V mutation. Light blue shaded area indicates the early outbreak period from October 7th to October 30th, 2021. Data were obtained from the GISAID database.

SARS-CoV-2 virus populations in this study were characterized by expansion and diversification over time. This finding is consistent with the long periods of infection seen in lions, and with the fundamental behavior of RNA viruses featuring rapid, but error-prone replication following the establishment of infection. Importantly, as a virus population expands, the potential strength of selection increases as well. Virus populations overall were evenly split between signatures of positive and purifying selection; however, all samples from hyenas demonstrated uniquely strong positive selection in comparison to the other two species. This difference could be related to the sample collection timeline, as hyena samples were collected later in the outbreak, and thus may have been collected later in their course of infection compared to samples from lions and tigers. Patterns of mutation and selection point to the possibility of an increased SARS-CoV-2 evolutionary rate in hyenas, as has been identified in white-tailed deer[47]. Our study was limited geographically and by the total number of animals in each collection. Future studies employing a larger sample size spanning a period of sustained within-species transmission, in combination with Bayesian phylogenetic techniques, would be needed to evaluate this hypothesis. Ultimately, within-host signatures of selection observed in our study represent potential directions of SARS-CoV-2 host species adaptation that could follow sustained transmission in a population of animals. It is important to note that our analyses

used a relatively small number of samples collected from tigers and hyena samples that may have been collected later in the course of infection than samples collected from lions. Regardless, the observed differences in overall nucleotide diversity could reflect a host-mediated species effect on mutation, as opposed to other aspects of the viral life cycle, like overall replication rate, assembly, or cell entry. As one possible example, APOBEC proteins vary across vertebrates, which may lead to host species-specific mutational signatures[48].

It is generally believed that following an initial period of evolution for transmissibility, the evolution of SARS-CoV-2 in human populations is driven by immune responses generated during vaccination and prior infection[43]. Though both of these drivers act at the between-host level on a global scale, the variation upon which selection acts is generated within individual hosts. Spillback from SARS-CoV-2-infected animals to humans has previously been reported across a wide range of species and environments, including zoo animals[33]. Here, no new, concerning variant lineage was detected, but a handful of potentially adaptive mutations became fixed in lions and hyenas, and one of these mutations may have subsequently spilled back into human populations. Thus, although SARS-CoV-2 infections in non-human animals may represent a fraction of global cases, they may have a disproportionate potential to contribute to virus genetic variation and evolution. The outbreak described in this study offered an exceptional case study to

witness the mechanisms underlying virus within-host evolution following host shifts in action.

## Methods

### Next-generation sequencing of qRT-PCR positive and inconclusive nasal swab samples

This study was approved by the Association of Zoos & Aquariums standardized research proposal process (Denver Zoo Conservation Alliance approval number 2023-11). Animals, sample collection, and diagnostic testing are described in Table S1 and previous published[16]. The sample size was predetermined by the number of individuals within each species housed at the zoo. Sex was not considered in the study design and analysis because the entire population of each species was included in sampling. During the time period of the outbreak (October 2021-January 2022), Denver Zoo was open to visitors while high-level biosecurity measures were in place. Virological controls included use of personal protective equipment by animal care staff, and assignment of different staff to care for each species. Animals were fed a variety of commercially sourced meat products from human grocery suppliers. Most commonly large felids were fed ground beef and venison; less frequently, they fed meat and bone products from pork, beef, bison, goat, rabbit, and chicken. Other diet components included whole prey (rabbits, guinea pigs, chicks, and quail), fish, eggs, canned cat foods, and produce that were commercially sourced.

Aliquots of all nasal swab samples that were positive or inconclusive by qRT-PCR[16] analysis were thawed and inactivated with Trizol reagent (Invitrogen) for five minutes at room temperature prior to RNA isolation using a modified protocol for the Zymo RNA Clean & Concentrator-5 kit with New England Biolabs DNase I. RNA was eluted in nuclease-free water, and kept on ice or stored at -80 °C before continuing with reverse transcription. Blank RNA extraction negative controls were also generated using nuclease-free water as input. Complementary DNA (cDNA) was generated using SuperScript IV Reverse Transcriptase (Invitrogen) and stored at -20 °C prior to tiled amplicon PCR. Complete protocols for RNA and cDNA are available in the project Github repository[49]. To control for false-positives resulting from PCR amplification, all biological samples were processed in two technical replicates for cDNA generation onwards. Replicate samples were enriched for SARS-CoV-2 genomic material using a tiled amplicon PCR protocol developed by the ARTIC Network. The ARTIC version 4 (V4) primer scheme was used to generate a pool of overlapping amplicons that cover the entire SARS-CoV-2 genome (Integrated DNA Technologies 10011442; Supplementary Data 7). Pooled PCR products were visualized on 1% agarose gels, and quantified using a Qubit dsDNA 1x High Sensitivity Assay kit (Thermo Fisher Scientific). All samples that amplified successfully and contained sufficient genomic material continued to library preparation. Libraries were prepared using 100 ng DNA as input using NEBNext Ultra II DNA Library Prep kits (New England Biolabs), and individual replicates were uniquely dual indexed with NEBNext Multiplex Oligos for Illumina (New England Biolabs). All bead cleanups were performed using Ampure XP beads (Beckman Coulter). Of the 68 total positive samples, we acquired high-quality sequencing libraries from 57 (84%) samples. Of the 47 total inconclusive samples, we acquired high-quality sequencing libraries from 6 (13%) samples.

All qRT-PCR positive samples were sequenced on the same Illumina MiSeq instrument at the Colorado State University Next Generation Sequencing Facility using v2 500 cycle 2 ×250 bp kits (Illumina). The technical replicate libraries for Lion F collected on October 21st were pooled and sequenced as a test run with a separate project. All other individual libraries from positive samples were pooled evenly into three sequencing libraries corresponding to three separate sequencing runs, each containing a blank RNA extraction negative control library that was subjected to all of the processing steps. All qRT-PCR inconclusive samples were sequenced on one run of

an Illumina MiSeq instrument at the University of Colorado Boulder Center for Microbial Exploration Sequencing Center using a v2 500 cycle 2 ×250 bp kit (Illumina). A spreadsheet detailing the samples and index sequences in each library is available in the project Github repository[49].

### Variant calling and bioinformatics

Sequencing data in FASTQ file format were input into the nf-core/viralrecon pipeline version 2.6.0 for detailed quality control metrics, lineage assignment, within-host viral variant calling, variant annotation, and the generation of consensus sequences. The pipeline is designed to perform low-frequency variant calling for viral samples. For initial analyses, we used the nf-core/viralrecon default as a SARS-CoV-2 reference sequence, GenBank MN908947.3. However, we used a modified GFF for MN908947.3 that separates *ORF1a* and *ORF1b* to account for the -1 ribosomal frameshift. This is consistent with the approach used by Nextstrain.

Briefly, nf-core/viralrecon merges paired FASTQ files and read quality is assessed (FastQC) prior to trimming for quality and adapters (fastp) and removal of host reads (Kraken 2). Reads are aligned to a reference sequence, and variant calling is performed (Bowtie 2, SAMtools, iVar, picard) followed by variant annotation (SnpEff, SnpSift). The pipeline uses the default iVar parameters including a minimum base quality of 20 and a minimum allele frequency of 0.03 for variant calling. Finally, consensus sequences are generated and assessed (BCFtools, QUAST, Pangolin, Nextclade). Coverage and quality metrics and a complete variant table, modified minimally for clarity, are available in Supplementary Data 3 and 4.

At this stage sequence data were subjected to a manual round of quality control prior to downstream analyses, using the metrics reported in the nf-core/viralrecon pipeline multiqc report (Supplementary Data 3). Replicate NGS datasets that had greater than or equal to 75% 1X and 10X coverage and less than 25,000 Ns per 100 kb were retained for within-host variant analysis. This removed three datasets generated from blank RNA extraction negative control samples, 16 datasets corresponding to both technical replicates for eight biological samples, and two single replicate datasets corresponding to two other biological samples (Lion I on November 22nd and Lion C on November 4th). After quality control, 124 NGS datasets corresponding to 63 biological samples from 16 individuals (three hyenas, 11 lions and two tigers) were retained. The mean sequencing depth of a dataset ranged from 2012X to 6997X. The overall mean and median sequencing depth of coverage across the full SARS-CoV-2 genome were 3754X and 2826X respectively.

For analysis requiring all consensus sequences in our dataset (the phylogenetic tree) one of the two technical replicates was selected using the quality control metrics generated by the nf-core/viralrecon pipeline. Of the two technical replicates, the sequence with a higher value for %coverage > 10x was selected. If this metric was the same, the sequence with fewer Ns per 100 kb was selected. If the number of Ns was the same, the sequence with a higher overall median coverage value was selected.

For analyses involving just one consensus sequence per individual (the haplotype network), we aimed to select the highest quality sequence collected at the latest date. To do this, the sequence generated at the latest timepoint with high coverage metrics (≥ 99 %coverage > 1X and ≥ 99 %coverage > 10X and <1225 Ns per 100kb) was selected. The dataset generated from the sample collected on Nov. 22nd from Hyena D passed the initial manual quality control step after NGS, but did not meet this more stringent cutoff. However, it was still included because it was the only sequence generated from this individual.

For a subset of additional analyses, a second bioinformatics pipeline for viral variant analysis was employed [https://github.com/stenglein-lab/viral_variant_caller]. This pipeline outputs raw

sequencing depth across all datasets at all genomic positions which we used to calculate summary statistics. It was also used to generate.vcf files in the appropriate format for input in the SNPGenie pipeline (see below).

Based on our conclusion that the two tigers were infected first, we also assessed within-host variation relative to what we refer to as the tiger reference sequence: the consensus SARS-CoV-2 sequence of the virus samples recovered from the two tigers tested on October 7th. The two tiger sequences were aligned with MUSCLE, and a consensus sequence was generated with emboss. This consensus was then used as the reference sequence for running the nf-core/viralrecon pipeline with a custom configuration, using the remaining NGS datasets from lions and hyenas (only those with two high quality technical replicates) as input.

## Mutation naming conventions and validation

For all mutations described and visualized in this report, the first naming convention used for mutations in the region that codes for the ORF1ab polyprotein is described as its position within either *ORF1a* or *ORF1b*. To help provide additional context, for mutations in this region we also list the specific nonstructural protein affected. We also interrogated sequencing coverage at the genomic position corresponding to each mutation discussed in this report to and confirmed >100X coverage in both sequencing replicates.

To identify mutations in our dataset that are characteristic of SARS-CoV-2 lineage AY.20, a list of 32 characteristic mutations was obtained from https://outbreak.info/situation-reports?pango=AY.20. This list is available as Supplementary Data 2. We modified this table with information about the nonstructural protein impacted by mutations in *ORF1ab*, and with alternate names for mutations based on the annotation and naming convention output by the nf-core/viralrecon pipeline.

Briefly, the outbreak.info mutation list includes E156G and del157/158 in *S*, whereas our dataset has the mutation E156_R158delinsG. This is the same change with a different name. Similarly, outbreak.info lists del119/120 in *ORF8*, whereas our dataset has D119_F120del. Finally, our dataset describes a C > T change at genomic position 27874, which would cause a T to an I amino acid change. In our annotation schema, this effect is described as upstream of *ORF8*, and in outbreak.info it is listed as T40I occurring in *ORF7b*. These are also the same change with a different name. Finally, S84L in *ORF8* is listed as a characteristic mutation of AY.20 in the outbreak.info list. However, this mutation was already present in the reference sequence used in our analysis (GenBank MN908947.3), and all sequences we generated also had the L residue at this position. Therefore, after accounting for these annotation disparities, all mutations characteristic of AY.20 were identified in our sequences.

Some mutations that appear in the Nextstrain phylogeny are not discussed in our results due to insufficient sequencing depth or failure to be detected in both technical replicates; these may represent sequencing errors.

## Public SARS2-CoV-2 data, phylogenetic trees and haplotype network

A time-based phylogenetic tree was generated with Nextstrain software following the ncov SARS-CoV-2 workflow[50]. All sequence data not generated in this study were obtained from GISAID[34]. The tree (Fig. 2A, B) included three data inputs: (1) sixty-three SARS-CoV-2 consensus sequences generated from zoo animals, (2) all human-derived sequences classified as AY.20 in the state of Colorado between September 23rd and November 4th, 2021 (N = 198 sequences), and (3) a random subsample of human-derived sequences in Colorado from the days leading up to the zoo outbreak (100 sequences/day from September 23rd to October 7th, 2021; N = 1500 sequences). The tree was visualized in Taxonium and

screenshots were used for Fig. 2A and B[51]. The interactive version is publicly available at https://nextstrain.org/community/laurabashor/DZSARS2/COAY20. A second tree (Figure S1) was generated with Nextstrain using the above method, including two data inputs: SARS-CoV-2 consensus sequences generated from the zoo outbreak (N = 63), and all publicly available sequences from animals classified as AY.20 (N = 3, two white-tailed deer and one cat). A third phylogenetic tree (Figure S2) was generated with Nextstrain using the above method, and including four data inputs: (1) sixteen SARS-CoV-2 consensus sequences from the Denver Zoo outbreak corresponding to the highest quality sequence collected at the latest date from each individual animal, (2) a random subsample of human-derived sequences in Colorado from the days leading up to the zoo outbreak (100 sequences/day from September 23rd to October 7th, 2021; N = 1500 sequences), (3) all felid- and hyena- derived SARS-CoV-2 sequences available in the GISAID database prior to December 2021, and (4) contextual sequences collected in the same time and place as each animal sequence (100 human-derived sequence per animal-derived sequence from the United States; 10 sequences for animal-derived sequences from other locations). Felids include domestic cats, lions, tigers, snow leopards, a fishing cat, and a leopard cat. The second tree is publicly available at https://nextstrain.org/community/laurabashor/DZSARS2/felidshumans. Detailed acknowledgements for all phylogenetic trees including the authors, originating and submitting laboratories of all sequence and metadata obtained from GISAID are available in GISAID EPI_SET_240407tz and EPI_SET_240407wv (see Supplementary Information). A haplotype network was generated with the 'pegas' package in R using an alignment (MUSCLE) of the latest high-quality sequence obtained from each individual.

SARS-CoV-2 genomes from nasal swab samples from all eleven lions collected on October 14th and one hyena collected on October 28th were also sequenced by USDA NVSL at the time of the outbreak (accessions included in GISAID EPI_SET_240407tz). A comparison of the consensus sequences obtained in our study from lions and hyenas on these dates reveals high overall identity, with two SNPs observed in the APHIS sequences that were not detected in our datasets. Based on shared SNPs, the hyena sequence (GISAID accession EPI_ISL_6810900) was likely generated from a sample from Hyena A. All other GISAID sequences discussed in this report are included in GISAID EPI_SET_240407tz. Unless otherwise specified in the text, all GISAID searches were carried out with the follow options selected: complete and low coverage excluded.

## Within-host virus population demographics and signatures of selection

Within-host virus population demographics and signatures of selection were calculated with SNPGenie[52]. The SNPGenie pipeline calculates nucleotide diversity as the mean number of pairwise differences per nonsynonymous or synonymous site in the genome from next-generation virus sequencing data, and estimates are weighted by allele frequencies at these sites. This analysis was done at both the population, or virus sample level, and by SARS-CoV-2 gene product.

Linear mixed models were used to evaluate statistical differences in π and πS over time (measured as days since first positive test), using the lmer() function in the 'lme4' package in R. Time was a fixed effect, with animal ID as a random effect to account for repeated measures of the same individual over time. Due to the limitation of the population size of the zoologic collection, which included only two tigers and four hyenas, we restricted our statistical analysis to lion samples in these models. The residuals of both models were normally distributed, and the statistical summary table was generated with the tab_model() function in the 'sjPlot' package in R. In this table, the conditional R2 for the πS model on the righthand side is NA because zero variance was explained by the animal ID grouping factor. No formal analysis was

done to distinguish endpoint $\pi$N/$\pi$S ratios for Fig. 5A due to insufficient sample size.

### Reporting summary

Further information on research design is available in the Nature Portfolio Reporting Summary linked to this article.

## Data availability

The next-generation sequencing datasets generated in this study have been deposited in the NCBI Sequence Read Archive under BioProject accession code PRJNA1154601. Processed data tables are provided in the Supplementary Information. All relevant raw data from each figure or table are available on Github at [https://doi.org/10.5281/zenodo.17350964]. Interactive versions and associated metadata for the three phylogenies are available at [https://nextstrain.org/community/laurabashor/DZSARS2]. GISAID sequences used in this study can be found in the Supplementary Information in GISAID EPI_SET_240407tz and EPI_SET_240407wv.

## Code availability

Code used for analyses and to generate figures and tables included in this study are available on Github at [https://doi.org/10.5281/zenodo.17350964].

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

## Acknowledgements

We gratefully acknowledge the many collaborators on this project including the dedicated care specialists and veterinary medical professionals who cared for these animals and coordinated sample collections, especially Jessica Long. Thank you to the staff of the CSU Veterinary Diagnostic Laboratories for diagnostic testing services. Thank you to Mary Nehring for laboratory management, and Dan Sloan, Greg Ebel and Toby Koch for insight and advice on data presentation and analysis. This research was approved by Denver Zoo Conservation Alliance through the Association of Zoos & Aquariums standardized research proposal process (Denver Zoo Conservation Alliance approval number 2023-11). We gratefully acknowledge all GISAID data contributors, i.e., the Authors and their Originating laboratories responsible for obtaining the specimens, and their Submitting laboratories for generating the genetic sequence and metadata and sharing via the GISAID Initiative. Research reported in this publication was supported by the National Institute Of Allergy And Infectious Diseases of the National Institutes of Health under Award Number T32AI162691 (LB). The content is solely the responsibility of the authors and does not necessarily represent the official views of the National Institutes of Health. A MARC Scholar was funded by a grant from the National Institute of General Medical Sciences of the National Institutes of Health: T34GM140958 (MG). Computational resources were supported by NIH/National Center for Advancing Translational Science Colorado Clinical and Translational Science Awards grant UL1 TR002535 (MDS). Research in this publication was also supported by the Colorado State University College of Veterinary Medicine and Biomedical Sciences Research Council Award (SV) and Colorado State University's Office of the Vice President for Research's Accelerating Innovations in Pandemic Disease initiative, made possible through support from The Anschutz Foundation (LB).

## Author contributions

LB and ENG – conceptualization, data curation, formal analysis, investigation, visualization, writing-original draft, writing-review & editing; MG, KE and KS – investigation, data curation, formal analysis, resources; LC, MS and JGJ – investigation, resources, writing-review & editing; SV and KP – conceptualization, funding acquisition, project administration, resources, supervision, writing-review & editing.

## Competing interests

The authors declare no competing interests.

## Additional information

**Peer review information** : *Nature Communications* thanks Dominik Melville, João Queirós and Vittorio Sarchese for their contribution to the peer review of this work. A peer review file is available.

