## [Transparent Peer Review file · Nature Communications]

SARS-CoV-2 within-host population expansion, diversification and adaptation in zoo tigers, lions and hyenas

Corresponding Author: Dr Sue Vandewoude

Version 0:

Reviewer comments:

Reviewer #1

(Remarks to the Author)

The manuscript entitled "SARS-CoV-2 within-host population expansion, diversification, and adaptation in zoo tigers, lions, and hyenas" by Bashor et al., provides a comprehensive analysis of SARS-CoV-2 genomic evolution and adaptation following a documented transmission event among non-domestic animal species, Amur tigers (*Panthera tigris altaica*), African lions (*Panthera leo*), and spotted hyenas (*Crocuta crocuta*) in a zoological setting. The investigation is methodologically robust, leveraging high-quality genomic sequencing and detailed bioinformatic and statistical analysis. The findings significantly advance our understanding of within-host SARS-CoV-2 evolution and species-specific adaptations. I recommend this manuscript for publication after addressing a few minor issues detailed below.

In the "Introduction" section, Authors can discuss in detail why zoological environments offer valuable contexts to study viral evolution, to further highlight the uniqueness and relevance of this research context. In the "Discussion" Authors should clearly outline the limitations of the study, including the relatively small sample size and limited geographic scope, to contextualize the generalizability of the findings. Additionally, authors could more explicitly suggest potential public health strategies for surveillance or biosecurity measures informed by the findings of this study.

Finally, I would like to ask the Authors, whether during the period under study, virological controls were also carried out on personnel responsible for managing these animals. This would be a very important aspect to evaluate to add evidence to the spillback and reverse zoonotic mechanisms that have characterized SARS-CoV-2.

Reviewer #2

(Remarks to the Author)

Dear Authors,

The manuscript entitled "SARS-CoV-2 within-host population expansion, diversification and adaptation in zoo tigers, lions and hyenas" represents an exceptional study of genomic surveillance of SARS-CoV-2 in 2 Amur tigers, 11 African lions and 4 spotted hyenas at the Denver Zoo, Colorado, USA. This study was conducted following an apparent spillover event from humans to tigers reported in previous studies, and was specifically designed to characterize the genomic evolution of the virus within the three host populations over a four-month period between October 2021 and January 2022. One of the main limitations of working with "non-experimental laboratory species" is often the uncertainty associated with uncontrolled environmental conditions and, in this case, the possibility of multiple viral transmissions and (re)infections from unknown sources. In my modest opinion, the phylogenetic trees provided with all SARS-CoV-2 delta AY.20 sublineages, including human and animal isolates (Figure 2a and 2b), and the network (Figure 2C) are not robust enough to support viral transmission from tigers to lions and hyenas. Despite the onset of symptoms in tigers, the lack of direct contact between the three zoo species, and the close phylogenetic position of the viruses isolated from lions and hyenas to those isolated from humans (gray lines), raises the question of the possibility of multiple infections from humans or other sources to lions and hyenas. The author's argument that "The species-specific clusters evident in this network are consistent with the observed onset of clinical signs, supporting the conclusion that SARS-CoV-2 infection began in tigers, then moved to lions, then to hyenas." is not robust, as networks do not provide directionality of transmission and only represent within-strain variability. There are other important questions: Was the zoo closed to humans during the study period? What is the food source of these animals? Are the human AY.20 cases found among zoo staff? What is the possible role of re-infection between

individuals? What is the phylogenetic relationship of the studied variants with the three other animal variants of AY.20 (2 white-tailed deer from New York and North Carolina in November 202; and 1 from a domestic cat in California in December 2021)? Are the mutations found in the study similar to those found in other animals? I suggest that the authors clarify some of these issues, if possible, and examine the data in the light of this evidence.

Best regards

Reviewer #3

(Remarks to the Author)

I have read the work by Bashor and colleagues. Its significance is encapsulated by this line:

L331: "The SARS-CoV-2 variation in lions and hyenas described in this study highlights the plasticity of within-host virus populations experiencing selection and genetic drift over time."

Given the work on white-tailed deer this finding is not novel per se. Is it interesting because it is on charismatic megafauna (albeit in zoos)? I won't judge that. But I wasn't convinced throughout that their research goes deep enough or is of general interest. In addition, the authors overstate some of their results concerning hyenas and tigers, which rest on very few samples, and an uneven sampling design that likely missed early phases of the infections (at the very least in hyenas).

Below I list a few of my concerns:

L93: How do you explain the positive tests weeks after the infection? Could you determine whether they are in the normal range of false positives?

Figure 2A and B I find difficult to read and visually not very convincing – is there maybe a nicer way to plot this than via nextstrain?

L210: why would an increase in synonymous nucleotide diversity over time matter, since it doesn't change AAs and functions?

Fig 4A: I am not convinced by a correlation consisting of 3 and 5 data point in tigers and hyenas respectively. I think the authors should focus on the lion results.

Fig 5. Are the authors sure that given the wide spread of the data in lions, the results from tigers and hyenas with such few samples are conclusively arguing for one or the other type of selection? Besides not all points are fully visible; on top I find it bizarre to compare all samples pairwise when some are taken early and some later in the infection. That surely has an impact on within host selection on the virus. Was purifying or positive selection more common among earlier or later samples?

L289: wouldn't that argue for tigers not to be the first link in the transmission chain? Could it not be independent transmissions? AY.20 with the mutation could be transmitted to the lions and hyenas while a different AY.20 without the mutation was transmitted to tigers. Additionally, hyenas were tested later. How can you even be sure they got it last?

L348: However, samples from hyenas were taken much latter than those from tigers and lions. You are likely missing the early infection period. I think this a strong assertion standing on shaky ground with few samples to back this up.

L355: I don't think this species effect on nucleotide diversity would hold up if more samples from all species existed and from early on in the infection. The tigers also have just 3 positive samples. Have you tested whether you have enough power to support your statistics?

Version 1:

Reviewer comments:

Reviewer #2

(Remarks to the Author)

Dear authors,

Thank you for your thorough and thoughtful revisions. You have addressed all my concerns as well as the other reviewer in a clear and satisfactory manner. The changes made have significantly improved the clarity and overall quality of the manuscript.

Reviewer #3

(Remarks to the Author)

This is the second time I am reviewing this manuscript and I am surprised at how few of the reasonable requests made by the reviewers have been addressed. In particular, reviewer 2's point regarding the uncertainty of transmission time line and the fact that the network analysis does not count as enough proof. I have raised a similar concern, and voiced that the uneven sampling design, particularly the fact that hyenas miss samples early on, leaves me unconvinced of the authors claim. Many responses were defensive and changes mainly concerned addendums to paragraphs without rethinking their approach.

Response to L93: Papers such as this manuscript have to stand on their own. Methods or results necessary to understand or strengthen the arguments should but included (even if only as shortened version) rather than citing another paper reporting similar findings and relying on the same data as the present one. If anything, their work (Gallichotte et al. 2025) brings into question how new the paper is they are trying to publish in Nature Communications now given that the take-home (i.e.,

supposed course of transmission from tigers to lions to hyenas) is already spelled out there (and even some graphics are strikingly similar e.g., Fig 4 in the msystems pub is Fig 1 in this MS). Generally, the practice of parsing out data into separate studies with only incremental knowledge gain is bad practice.

Response to Fig 4A: Justification based on the fact that someone else has done something does not make the statistics sound. Given the variability clearly visible in the data points in Figure 4A I don't understand why the authors seem to think that the hyena's correlation is meaningful. To spell it out: three of the 5 data points are at day 0 (somewhat clustered), the fourth is lower, whereas the fifth is super high, pulling the correlation into the shape it has, and let's not talk about the three data points for tigers.

In terms of sample sizes, I appreciate that the authors are limited with that respect and I do acknowledge the dense virological surveillance data they were able to collect. Still, given Figure 1, I believe that had to be a conversation on what samples were included for the subsequent analysis. Apart from the samples collected close to the outbreak, the selection seems random. Here I would have appreciated a more thought through approach or is their approach and final sample sizes down to what the authors describe in L450 and following?

Version 2:

Reviewer comments:

Reviewer #3

(Remarks to the Author)

I thank the authors for their patience with the reviewing process. I do think the manuscript has improved from its original version thanks to the contributions of all reviewers. I am satisfied with the revisions layed out before me in this last round. The authors recognise the shortcomings of their data sufficiently. I wish you much success with the final editorial process and future work in the field.

Response to Reviews

We thank the reviewers for their thoughtful reading of, and comments on our manuscript. We have provided a point-by-point response to their comments below.

Reviewer #1

The manuscript entitled “SARS-CoV-2 within-host population expansion, diversification, and adaptation in zoo tigers, lions, and hyenas” by Bashor et al., provides a comprehensive analysis of SARS-CoV-2 genomic evolution and adaptation following a documented transmission event among non-domestic animal species, Amur tigers (*Panthera tigris altaica*), African lions (*Panthera leo*), and spotted hyenas (*Crocuta crocuta*) in a zoological setting. The investigation is methodologically robust, leveraging high-quality genomic sequencing and detailed bioinformatic and statistical analysis. The findings significantly advance our understanding of within-host SARS-CoV-2 evolution and species-specific adaptations. I recommend this manuscript for publication after addressing a few minor issues detailed below.

We thank the reviewer for their thoughtful and positive assessment of our work.

In the “Introduction” section, Authors can discuss in detail why zoological environments offer valuable contexts to study viral evolution, to further highlight the uniqueness and relevance of this research context.

We thank the reviewer for pointing out the uniqueness and valuable context offered by our research collaboration with a zoo. We have added text to the introduction (lines 66-68) and discussion (lines 315-321) highlighting how this setting offered a unique opportunity for studying viral evolution.

In the “Discussion” Authors should clearly outline the limitations of the study, including the relatively small sample size and limited geographic scope, to contextualize the generalizability of the findings.

We have modified the text of the discussion to clearly outline these limitations as suggested (lines 381-382 and 388-391).

Additionally, authors could more explicitly suggest potential public health strategies for surveillance or biosecurity measures informed by the findings of this study.

We have modified the text of the discussion (lines 315-321) to discuss public health strategies and reference the companion article Gallichotte et al. 2025 (<https://doi.org/10.1128/msphere.00989-24>) in which these topics are discussed more comprehensively.

Finally, I would like to ask the Authors, whether during the period under study, virological controls were also carried out on personnel responsible for managing these animals. This would be a very important

aspect to evaluate to add evidence to the spillback and reverse zoonotic mechanisms that have characterized SARS-CoV-2.

The virological controls carried out by zoo personnel are described fully in a companion article describing this outbreak (Gallichotte et al. 2025 <https://doi.org/10.1128/msphere.00989-24>). We updated this citation (was previously cited as a preprint), in the references section (lines 639-640), and added additional information about zoo virological controls to the methods section (lines 412-414).

Reviewer #2

Dear Authors,

The manuscript entitled "SARS-CoV-2 within-host population expansion, diversification and adaptation in zoo tigers, lions and hyenas" represents an exceptional study of genomic surveillance of SARS-CoV-2 in 2 Amur tigers, 11 African lions and 4 spotted hyenas at the Denver Zoo, Colorado, USA. This study was conducted following an apparent spillover event from humans to tigers reported in previous studies, and was specifically designed to characterize the genomic evolution of the virus within the three host populations over a four-month period between October 2021 and January 2022.

We thank the reviewer for their thoughtful comments, and appreciate that our manuscript “represents an exceptional study of genomic surveillance”.

One of the main limitations of working with "non-experimental laboratory species" is often the uncertainty associated with uncontrolled environmental conditions and, in this case, the possibility of multiple viral transmissions and (re)infections from unknown sources. In my modest opinion, the phylogenetic trees provided with all SARS-CoV-2 delta AY.20 sublineages, including human and animal isolates (Figure 2a and 2b), and the network (Figure 2C) are not robust enough to support viral transmission from tigers to lions and hyenas. Despite the onset of symptoms in tigers, the lack of direct contact between the three zoo species, and the close phylogenetic position of the viruses isolated from lions and hyenas to those isolated from humans (gray lines), raises the question of the possibility of multiple infections from humans or other sources to lions and hyenas.

We've updated the language of the results (lines 137, 140-142) and discussion (lines 305-306) to address these limitations and the possibility of multiple infections. We also note that this reviewer's suggestion to evaluate additional other AY.20 genotypes (addressed below) illustrates the high identity among the Denver Zoo isolates relative to other animal-derived AY.20 isolates. This further supports a single infection event that subsequently spread within the zoo.

The author's argument that “The species-specific clusters evident in this network are consistent with the observed onset of clinical signs, supporting the conclusion that SARS-CoV-2 infection began in tigers, then moved to lions, then to hyenas.” is not robust, as networks do not provide directionality of transmission and only represent within-strain variability.

We've revised this statement in the results and clarified that analysis cannot determine directionality (lines 140-142).

There are other important questions: Was the zoo closed to humans during the study period? What is the food source of these animals? Are the human AY.20 cases found among zoo staff?

We have added additional information in the methods noting that the zoo was open to humans during the time period and the source of food for the animals (lines 412-420). Due to protections provided in the 1996 Health Insurance Portability and Accountability Act (HIPAA), we are unable to identify specific individuals in Colorado who were infected with AY.20.

What is the possible role of re-infection between individuals?

Based on our viral genomic analysis, we are unable to distinguish between recrudescence and reinfection and have noted this on lines 122-125.

What is the phylogenetic relationship of the studied variants with the three other animal variants of AY.20 (2 white-tailed deer from New York and North Carolina in November 202; and 1 from a domestic cat in California in December 2021)? Are the mutations found in the study similar to those found in other animals? I suggest that the authors clarify some of these issues, if possible, and examine the data in the light of this evidence.

We have generated a new supplemental figure that incorporates the three other animal variants of AY.20 and our zoo sequences (Figure S1 in the supplementary information, also publicly available at <https://nextstrain.org/community/laurabashor/DZSARS2/AY20animals>). These three animal variants of AY.20 were distinct from those seen at the zoo, providing further support for our hypothesis that this outbreak was caused by a single introduction. We discuss this on lines 133-136 of the results. We have updated all text that re-designates Figure S1 as Figure S2 in this revision and revised: (1) the methods (lines 554-558, 570; (2) the data availability statement (line 765), and (3), the supplementary information (SI lines 35, 76-82).

Reviewer #3

I have read the work by Bashor and colleagues. Its significance is encapsulated by this line: L331: "The SARS-CoV-2 variation in lions and hyenas described in this study highlights the plasticity of within-host virus populations experiencing selection and genetic drift over time." Given the work on white-tailed deer this finding is not novel per se. Is it interesting because it is on charismatic megafauna (albeit in zoos)? I won't judge that. But I wasn't convinced throughout that their research goes deep enough or is of general interest. In addition, the authors overstate some of their results concerning hyenas and tigers, which rest on very few samples, and an uneven sampling design that likely missed early phases of the infections (at the very least in hyenas).

We thank the reviewer for the comprehensive reading and assessment of our manuscript. We have revised the text throughout to highlight the limitations of our study and to address concerns relating to novelty of these findings.

Below I list a few of my concerns:

L93: How do you explain the positive tests weeks after the infection? Could you determine whether they are in the normal range of false positives?

Our companion article describing this outbreak (Gallichotte et al. 2025 <https://doi.org/10.1128/msphere.00989-24>) describes the comprehensive scheme for determining positive, negative and inconclusive results, demonstrates recovery of infectious virus in a subset of those positive samples, and discusses this in the broader context of SARS-CoV-2 rebound infections. We think that this publication addresses this comment, and accordingly have included the reference to the testing result classification to the methods section (lines 419-420).

Figure 2A and B I find difficult to read and visually not very convincing – is there maybe a nicer way to plot this than via nextstrain?

Thank you for this helpful comment. We've updated figures 2A and B to enhance the readability and impact of this figure. Specifically, we have made the following modifications: (1) we have visualized the Figure 2 phylogenetic tree in Taxonium to facilitate interpretation; (2) we have increased text size, removed gridlines, and highlighted external nodes; (3) date labels in subfigure 2D have been updated in BioRender to be consistent with subfigures 2A and B; (4) the figure legend (lines 161 and 168-169) and (5) the methods (lines 552-553) have been updated to reflect these changes.

L210: why would an increase in synonymous nucleotide diversity over time matter, since it doesn't change AAs and functions?

We've added to the discussion highlighting the importance of genetic diversity for viral evolution, and the potential significance of this diversity outside of functional changes (e.g, alternating codon usage, RNA secondary structure, innate immune evasion, etc.) to address this point (lines 366-369).

Fig 4A: I am not convinced by a correlation consisting of 3 and 5 data point in tigers and hyenas respectively. I think the authors should focus on the lion results.

We agree with the reviewer that a larger sample size for tigers and hyenas would have been preferred but also acknowledge that this study represents a unique number of sequentially sampled naturally infected individuals compared to similar studies in wildlife (see recent Nature Communications article: <https://doi.org/10.1038/s41467-024-53766-5> for one example). We have modified the text to more clearly take note of how sample size limits our conclusions

based on species in the results (line 220), discussion (lines 381-382 and 388-391) and methods (lines 596-598).

Fig 5. Are the authors sure that given the widespread of the data in lions, the results from tigers and hyenas with such few samples are conclusively arguing for one or the other type of selection? Besides not all points are fully visible; on top I find it bizarre to compare all samples pairwise when some are taken early and some later in the infection. That surely has an impact on within-host selection on the virus.

In order to address this concern, we have revised the text of the figure legend and results associated with Figure 5 regarding conclusions about type of selection (lines 227-229), interspecific sample comparisons (lines 234-236), and results that address whether purifying or positive selection was most common among earlier or later samples (lines 229-230). We have also described how to interpret points for which the $\pi N/\pi S$ ratio is infinite in the figure legend (lines 271-273).

L289: wouldn't that argue for tigers not to be the first link in the transmission chain? Could it not be independent transmissions? AY.20 with the mutation could be transmitted to the lions and hyenas while a different AY.20 without the mutation was transmitted to tigers. Additionally, hyenas were tested later. How can you even be sure they got it last?

Clinical signs reported support tigers as the initially infected species, and the genomic data presented support this conclusion. We have, however, revised text in the results and legends to note that it is possible that multiple independent transmission events occurred (lines 140-142 and 305-306).

L348: However, samples from hyenas were taken much later than those from tigers and lions. You are likely missing the early infection period. I think this is a strong assertion standing on shaky ground with few samples to back this up.

We appreciate the reviewer's caution to avoid overstating our findings, and have revised the text to note this limitation (lines 376-379).

L355: I don't think this species effect on nucleotide diversity would hold up if more samples from all species existed and from early on in the infection. The tigers also have just 3 positive samples. Have you tested whether you have enough power to support your statistics?

Although a larger sample size would have increased our power and been ideal, the sample size was predetermined by the number of individuals within each species housed at the zoo (note that the entire population of each species was included in sampling). The longitudinal sampling protocol providing repeated screening and defined the period that viral nucleic acid sequences could be detected as reported in Gallichotte et al. 2025 (<https://doi.org/10.1128/msphere.00989-24>). While overall population size is small, these sampling data are unprecedented in natural

disease outbreaks and provide a comparatively comprehensive analysis of a natural viral outbreak. Further, our statistical models took sample size into account when comparing among species. We have added additional text to the results (line 220), the discussion (lines 381-382 and 388-391) and the methods (lines 596-598) to clearly note sample size limitations. We did not carry out a power calculation before collecting samples or to plan an experimental design because samples were collected for surveillance purposes, and all animals within an exhibit were sampled.

Response to Reviews

Thank you very much for the opportunity to revise our manuscript, “SARS-CoV-2 within-host population expansion, diversification and adaptation in zoo tigers, lions and hyenas,” for Nature Communications. We have provided responses to each of the Reviewers’ queries.

Reviewer comments:

Reviewer #2 (Remarks to the Author):

Dear authors,

Thank you for your thorough and thoughtful revisions. You have addressed all my concerns as well as the other reviewer in a clear and satisfactory manner. The changes made have significantly improved the clarity and overall quality of the manuscript.

We thank the reviewer for their reviewing efforts, positive comments and support of the revised manuscript.

Reviewer #3 (Remarks to the Author):

This is the second time I am reviewing this manuscript and I am surprised at how few of the reasonable requests made by the reviewers have been addressed. In particular, reviewer 2’s point regarding the uncertainty of transmission time line and the fact that the network analysis does not count as enough proof. I have raised a similar concern, and voiced that the uneven sampling design, particularly the fact that hyenas miss samples early on, leaves me unconvinced of the authors claim. Many responses were defensive and changes mainly concerned addendums to paragraphs without rethinking their approach.

In our previous revisions we addressed the uneven sampling design (which was conducted by zoologic staff as part of a rigorous diagnostic and biosecurity protocol, not as an experimental design), and added additional text highlighting these limitations (lines 380-381, 385-388). We also addressed Reviewer #2’s point regarding the uncertainty of the transmission timeline by updating the language of the results (lines 141-143) and discussion (lines 304-305) to clarify that our analysis cannot determine directionality and address the possibility of multiple infections. As noted above, Reviewer #2 was satisfied with these revisions.

Response to L93: Papers such as this manuscript have to stand on their own. Methods or results necessary to understand or strengthen the arguments should but included (even if only as shortened version) rather than citing another paper reporting similar findings and relying on the same data as the present one.

In our previous revision, we added the following text to the results on lines 123-126 in our point-by-point response to Reviewer #2 concerns. We failed to point to this revised text in our previous point-by-point response to Reviewer #3', although we feel that it directly responds to their query in their initial review:

Lines 123-126:

Due to limited genetic variability across all sequences, we are unable to conclusively determine if positive samples detected in lions in November and December were recrudescence infections or reinfections; however, this homogeneity is suggestive of recrudescence infections.

If anything, their work (Gallichotte et al. 2025) brings into question how new the paper is they are trying to publish in Nature Communications now given that the take-home (i.e., supposed course of transmission from tigers to lions to hyenas) is already spelled out there (and even some graphics are strikingly similar e.g., Fig 4 in the mSystems pub is Fig 1 in this MS). Generally, the practice of parsing out data into separate studies with only incremental knowledge gain is bad practice.

Gallichotte et al. 2025 provides an epidemiological description of the outbreak in the context of other SARS2 outbreaks in conservatory animals, but does not describe the order of transmission between the animals, viral evolution or viral genomic analysis within and among the different species. Multiple publications describing different aspects of complex disease outbreaks is a common practice. (Relevant example: SARS2 at the Bronx Zoo, <https://pmc.ncbi.nlm.nih.gov/articles/PMC7554670/> and <https://pubmed.ncbi.nlm.nih.gov/33480553/>).

While Figure 4 in Gallichotte et al and Figure 1 in our manuscript are similar, they are used to illustrate entirely different points. Figure 4 in the mSphere paper demonstrates the result of the vRNA testing and PCR diagnosis of animals over time. Figure 1 in this manuscript demonstrates the outcome/success rate of next-generation sequencing attempts of PCR positive and inconclusive samples, highlighting which samples were deep-sequenced for the subsequent analyses reported. Providing Figure 1 in this paper allows readers to obtain critical information needed to understand sampling strategy and origin and temporal relationship of viral samples that were interrogated in this study.

As noted below in our comments in response to sample sizes, we have updated the results text for (lines 103-105) Figure 1 and its figure legend (lines 106-114) to emphasize the findings presented in this figure, specifically, that all of these samples were subjected to an NGS workflow, and which samples produced high-quality sequence data (all of which were included in genomic analysis). We have modified Figure 1 by adding a more distinctive, wider diamond-shaped border to each sample that was successfully sequenced with high quality resolvable data for further analysis. We also updated the text of the methods (lines 407-450) to further clarify.

Although we appreciate the reviewer's stated concern, we believe this manuscript stands alone and presents a body of novel genomic sequencing data generated that has provided novel and important findings about natural SARS-CoV2 disease cross-species transmission, and within and between host and species evolution.

Response to Fig 4A: Justification based on the fact that someone else has done something does not make the statistics sound. Given the variability clearly visible in the data points in Figure 4A I don't understand why the authors seem to think that the hyena's correlation is meaningful. To spell it out: three of the 5 data points are at day 0 (somewhat clustered), the fourth is lower, whereas the fifth is super high, pulling the correlation into the shape it has, and let's not talk about the three data points for tigers.

We're grateful for the reviewer's persistence in helping us improve this part of the analysis. We've considered the dataset and we can see how although our model was statistically significant, the biological significance is unclear given how little data was available for hyenas and tigers. We therefore modified our statistical models to evaluate only the lion sample (N=54 samples from 11 different individuals) dataset. We have subsequently generated a new Figure 4 (lines 240-246). that visualizes the data from lions separately, with the lions now shown in panel A, and tigers and hyenas in panel B. We revised the results and discussion text (lines 219-222, 385-388) and updated Table 1 accordingly (lines 248-254). Finally, we updated the methods section (lines 585-588) to provide complete details of this analysis.

Excerpted revised figure and table legends and results text:

Lines 240-246:

Figure 4. SARS-CoV-2 within-host populations underwent expansion and diversification over time. Each point indicates the nucleotide diversity of a SARS-CoV-2 virus population observed in one sample, calculated as the mean of two sequencing replicates. **(A, B)** Within-host nucleotide diversity (π) by the number of days after an individual's first positive SARS-CoV-2 test, in **(A)** lions (N=54 samples) and **(B)** tigers (N=3 samples) and hyenas (N=6 samples). **(C)** Within-host synonymous nucleotide diversity (π_S) over time by individual animal.

Lines 248-254:

Table 1. Summary of linear mixed models of nucleotide diversity (π) of SARS-CoV-2 in lions as a function of time. Time was measured as days since first positive qRT-PCR test. Time was a fixed effect, with animal ID as a random effect to account for repeated measures of the same individual (N=54 samples from 11 lions) over time. The model was fit with the lmer() function in the 'lme4' package and this table was generated with the tab_model() function in the 'sjPlot' package in R.

Lines 219-222:

To characterize selective pressures on within-host SARS-CoV-2 populations, we calculated nucleotide diversity (ρ) across the virus genome at nonsynonymous and synonymous sites (**Tables S6 and S7, Figure 4A, B**). In lions, which were sampled repeatedly over time, we observed a significant increase in within-host

nucleotide diversity over time, accounting for repeated measures of the same individuals (**Figure 4A, Table 1**). Within individual lions, synonymous nucleotide diversity also increased over time (**Figure 4C, Table 1**).

In terms of sample sizes, I appreciate that the authors are limited with that respect and I do acknowledge the dense virological surveillance data they were able to collect. Still, given Figure 1, I believe that had to be a conversation on what samples were included for the subsequent analysis. Apart from the samples collected close to the outbreak, the selection seems random. Here I would have appreciated a more thought through approach or is their approach and final sample sizes down to what the authors describe in L450 and following?

We are thankful to the reviewer for drawing our attention to the need to clarify the description of the sample selection process. In this study we attempted to sequence all the positive (red, N=68) and inconclusive (yellow, N=47) samples shown in Figure 1, which represent all the samples collected and tested as positive or inconclusive during the outbreak. We were able to successfully generate high-quality NGS data from 63 samples outlined in blue in Figure 1 (57 positives, 6 inconclusives).

As noted above, we have revised Figure 1 and its figure legend (lines 106-114) to emphasize that this is the rationale for providing this figure. We added a more distinctive, wider diamond-shaped border to each sample that yielded high-quality genomic sequencing libraries.

We have also updated the results text (lines 103-105) accordingly. We also modified and reorganized the methods section (lines 407-450) to describe this approach. Specifically, we added clarifying text, and combined the two sections describing positive and inconclusive sample library preparation into one section to clarify that all the samples with positive or inconclusive PCR results were subjected to the same workflow.

We thank the reviewers for their helpful suggestions and hope that these clarifications and modifications address these outstanding concerns.

Excerpted revised text from manuscript:

Lines 103-105:

Of the 68 total qRT-PCR-positive swab samples shown in Figure 1, 57 (84%) yielded high-quality sequencing datasets. Of the 47 qRT-PCR-inconclusive samples, we acquired high-quality sequencing datasets from 6 (13%) samples.

Lines 106-114:

Figure 1. SARS-CoV-2 RNA was persistently detected in tigers, lions and hyenas by qRT-PCR followed by next-generation sequencing. qRT-PCR was performed on nasal swabs from tigers (N=2), lions (N=11) and hyenas (N=4) sampled repeatedly between October 2021 and January 2022. Each point represents a nasal swab sample tested for three SARS-CoV-2 gene targets,

colored by the result determined by diagnostic software. All the positive (red, N=68) and inconclusive (yellow, N=47) samples shown were subjected to the NGS library preparation workflow. Points with a blue diamond border indicate samples from which high-quality NGS data was successfully generated (N=57 positive and N=6 inconclusive samples, N=63 total).

Line 407:

Next-generation sequencing of qRT-PCR positive and inconclusive nasal swab samples

Line 417:

Aliquots of all nasal swab samples that tested positive or inconclusive by qRT-PCR...

Lines 432-434:

All samples that amplified successfully and contained sufficient genomic material continued to library preparation.